# Correlation study of FGF23/D-serine in maintenance hemodialysis patients with combined hearing impairment

Dunlu Yuan[1], Jiaqing Li[1], Min Guo[2], Qing Yang[1], Jingjing Huang[3], Jingwen Nie[1], Ruomei Li[2], Qing Li [ID][1] *

1 Department of Nephrology, The First Affiliated Hospital of Kunming Medical University, Kunming, China,
2 Department of Otolaryngology, The First Affiliated Hospital of Kunming Medical University, Kunming, China, 3 Department of Medical Record, The Third People's Hospital of Kunming, Kunming, China

* liqing@kmmu.edu.cn

**Data Availability Statement:** All relevant data are within the manuscript and its Supporting Information files.

## Abstract

### Background

Recent studies have reported an association between chronic renal failure and hearing impairment. Yet, the exact mechanism of action is still not fully understood. In this study, we investigated the expression of fibroblast growth factor 23 (FGF23) and D-serine in maintenance hemodialysis (MHD) patients with end-stage renal disease (ESRD) complicated with hearing impairment and further investigated the correlation between FGF23/D-serine and hearing impairment.

### Methods

A total of 90 subjects, including 30 MHD patients complicated with hearing impairment, 30 MHD patients with normal hearing, and 30 controls, were included in this case-control study. Relevant data were obtained by questionnaire survey, audiometric test, enzyme-linked immunosorbent assay (ELISA) to determine FGF23 level, and high-performance liquid chromatography to determine D-serine level.

### Results

MHD patients showed abnormally high expression of FGF23 and D-serine, where FGF23 and D-serine levels were significantly higher in the group with hearing impairment than in the group with normal hearing and normal controls (all P<0.01). Also, elevated FGF23 and D-serine were identified as risk factors for hearing impairment in ESRD, with ORs of 16.54 (95%CI, 2.75–99.55) and 15.22 (95%CI, 2.59–89.51), respectively. Further Person correlation analysis showed a moderate positive correlation between FGF23 and D-serine (r = 0.683, P<0.001).

### Conclusion

This study provides potential biomarkers for the early detection of hearing impairment complicated by chronic renal failure, and the reduction of FGF23/D-serine may provide a

**Funding:** This study was supported by grants from the Major Teaching Educational Reform Project of Kunming Medical University (2022-JY-Z-12), the Project of College Education Cooperation Project of Yunnan Province (SYSX202034), the Subproject of Fund of CKD Clinical Research Center in Yunnan Province (202102AA100060), the Project of Yunnan health training project of high level talents (H-2018052), and Hundred of young and middle-aged academic and technical talent in Kunming Medical University (60118260103). The funders had no role in study design, data collection and analysis, decision to publish, or preparation of the manuscript.

**Competing interests:** The authors have declared that no competing interests exist.

potential target for the treatment of hearing impairment complicated by chronic renal failure.

## Introduction

Chronic kidney disease (CKD) is a global health problem that affects more than 13% of the general population worldwide [1]. It is characterized by a possible progressive decline in kidney function leading to renal failure, which can eventually progress to end-stage renal disease (ESRD) or uremia. Moreover, recent studies have found that the incidence of sensorineural deafness in patients with chronic renal failure ranges from 40% to 90%, and the degree of deafness is positively correlated with the degree of renal failure [2, 3]. Hearing loss is the fifth leading cause of disability in the world [4]. It is one of the risk factors for cognitive decline that is independently associated with dementia [5, 6]. Hearing impairment in chronic renal failure leads to increased disability, further loss of quality of life, and increased disease burden in patients with renal failure [7].

Fibroblast growth factor-23 (FGF23), discovered by Yamashita *et al.* in 2000, is a regulatory factor produced and secreted by osteoblasts and osteofibroblasts, which has an important role in the regulation of serum phosphorus, parathyroid hormone (PTH) and $1,25\text{-}(OH)_2\text{-}Vit\text{-}D_3$ [8]. Previous studies have found that CKD patients have elevated FGF23, and this increase appears to precede the increase in other serum parameters, including creatinine, urea nitrogen, and PTH [9]. Thus, elevated FGF23 is considered a sensitive biomarker for renal and extrarenal adverse effects in patients with CKD [10]. Furthermore, abnormally high expression of FGF23 in patients with chronic renal failure accelerates a series of extrarenal damages such as organismal calciphylaxis, atherosclerosis, secondary bone disease, and cognitive impairment of the nervous system [11–13].

D-serine is a non-essential amino acid involved in various physiological activities and pathological processes, including excitatory neurotransmission. Endogenous D-serine acts as a neurotransmitter released by neurons and plays an important regulatory role in neurodevelopment, neurotoxicity, learning, and memory [14]. Studies have also shown that chronic renal failure leads to abnormal amino acid metabolism and elevated levels of D-serine in the peripheral blood of patients with chronic renal failure, which is associated with disease progression [15].

In this study, we investigated the correlation between FGF23/D-serine and hearing impairment in MHD patients with ESRD, which is expected to provide a theoretical basis for the treatment of hearing impairment induced by chronic renal failure and a potential target for drug therapy.

## Material and methods

### Research subjects

The ethical review of the First Affiliated Hospital of Kunming Medical University approved the study, and consent was recorded via a paper form. Participants were presented with some information about this study followed by a consent statement: "I have read this Informed Consent. I have clearly understood the relevant content, and voluntarily cooperate to complete the questionnaire, physical examination and sampling work." All enrolled participants signed the informed consent form.

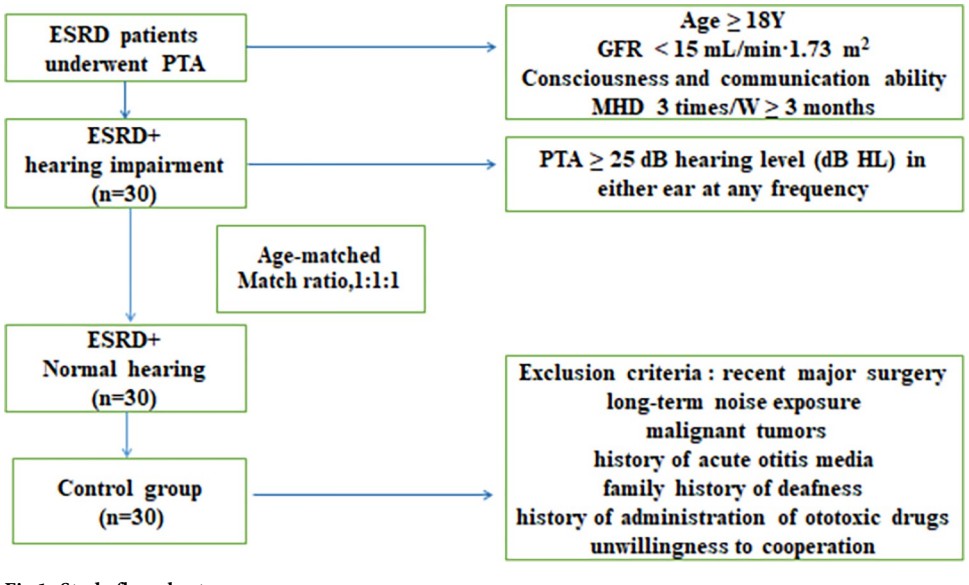

**Fig 1. Study flow chart.**

Patients who underwent hearing tests by pure tone audiometry (PTA) during maintenance hemodialysis at the First Affiliated Hospital of Kunming Medical University from May to July 2022 were enrolled in this study. First, 30 MHD patients with ESRD complicated by hearing impairment were selected, and then 30 MHD patients with normal hearing and 30 healthy controls were matched 1:1 by age for the case-control study (Fig 1).

## Inclusion and exclusion criteria for ESRD patients

Inclusion criteria were: age $\geq$ 18 years; glomerular filtration rate (GFR) < 15 mL/min-1.73 m$^2$; receiving regular hemodialysis for more than 3 months, four hours per session, 3 times/week; conscious; willing to participate in the study.

Exclusion criteria were: age < 18 years; in the induction phase of hemodialysis; history of long-term noise exposure or ototoxic drug use; history of acute otitis media; family history of deafness; history of recent surgery; malignancy; unwillingness to cooperate with the study; history of severe mental illness.

## Inclusion and exclusion criteria for the control group

Inclusion criteria were: age $\geq$ 18 years; no previous history of renal disease; normal audiometry; conscious and normal communication; willingness to participate in the study.

Exclusion criteria were: age < 18 years; previous history of renal disease; history of chronic noise exposure or ototoxic drug use; history of acute otitis media; family history of deafness; unwillingness to cooperate.

## Hearing impairment definition

A Conera pure tone audiometer (MADSEN Co., Ltd., Copenhagen, Denmark) was used to measure at frequencies of 250, 500, 1000, 2000, 4000, and 8000, respectively, and monaural pure tone audiometry (PTA) of $\geqq$ 25 dB at any one frequency was considered as hearing impairment [16].

## Data collection

**Questionnaire survey.** A self-designed questionnaire form was used, which included questions regarding age, gender, marital status, education, history of previous diseases, history of ear disease, history of kidney disease, complications of kidney disease, etc. The respondents' height, weight, and BMI were also analyzed. We had access to information that could identify individual participants during or after data collection.

**Hearing tests.** Binaural PTA was performed using a pure tone audiometer (MADSEN Co., Ltd., Copenhagen, Denmark) at frequencies of 250, 500, 1000, 2000, 4000, and 8000 in a dedicated audiometric room to obtain PTA values for each frequency.

**Laboratory data.** Laboratory data included blood creatinine, blood urea nitrogen (BUN), albumin (ALB), uric acid (UA), blood calcium, blood phosphorus, blood potassium, PTH, Hemoglobin (Hb), and red blood cell count (RBC).

**Measurement of peripheral blood FGF23 and D-serine levels.** All subjects fasted for at least 8 h. Then, 5 ml of venous blood was collected before 8:30 am. To avoid the influence of hemodialysis on laboratory data in MHD patients, blood was not collected during and 1 day after hemodialysis. Sampling for laboratory examination was carried out one-time.

The supernatant was extracted by centrifugation at 3000 rpm/min for 10 min. The levels of FGF23 in serum were measured using a commercially available ELISA assay kit (Elabscience Co., Wuhan, China) according to the manufacturer's instructions. In addition, serum D-serine levels were measured at 535/587 nm endpoint mode using a micro 2D-HPLC platform in combination with a multi-well fluorescence enzyme marker according to the D-serine assay reagent instructions.

## Statistical analysis

EpiData 3.1 software was used to establish the database, and SPSS 22.0 (IBM, Armonk, NY, USA) was used for statistical analysis. The categorical variables were compared as ratios or rates using the chi-square test; continuous variables were described using "Mean±SD," and differences between groups were analyzed using the F-test; dichotomous logistic regression was used to analyze the factors influencing hearing impairment in ESRD patients. The Person correlation coefficient was used to describe the correlation between continuous variables. The test level of bilateral α was 0.05, and $P < 0.05$ was considered statistically significant.

## Results

A total of 90 subjects were selected for this study, including 30 MHD patients complicated by hearing impairment, 30 MHD patients with normal hearing, and 30 controls. As seen in **Table 1**, the two MHD groups had a higher prevalence of hypertension, blood creatinine, BUN, PTH, blood phosphorus, FGF23, and D-serine than the control group, but also lower Hb, RBC, and ALB (all $P < 0.01$). Furthermore, among the three study groups, the highest levels of FGF23 and D-serine were found in the MHD group with hearing impairment, followed by the group with normal hearing in MHD (all $P < 0.01$) (**Figs 2** and **3**).

### Analysis of factors influencing hearing impairment in patients with ESRD

The presence of hearing impairment in MHD patients was a dependent variable (0 = none, 1 = yes), while gender (0 = male, 1 = female), diabetes mellitus (0 = none, 1 = yes), hyperuricemia (0 = none, 1 = yes), anemia (0 = none, 1 = yes), hypoalbuminemia (0 = none, 1 = yes), blood potassium (0<5.3mmo/l, 1≧5.3mmo/l), blood calcium (0<2.52mmol/l, 1≧2.52mmol/l), blood phosphorus (0<1.51mmol/l, 1≧1.51mmol/l), PTH (0<300pg/ml, 1≧300pg/ml),

**Table 1. Basic information.**

| Characteristic | Control group (n = 30) | MHD+NH group (n = 30) | MHD+HI group (n = 30) | P |
|---|---|---|---|---|
| Age(y) mean(SD) | 43.17±8.95 | 43.43±8.79 | 43.90±10.38 | 0.954 |
| Male no. (%) | 16(53.3) | 15(50.0) | 22(73.3) | 0.139 |
| BMI (Kg/m$^2$) mean(SD) | 23.96±3.19 | 22.30±3.90 | 22.53±3.61 | 0.156 |
| History of hypertension no. (%)** | 12(40.0) | 30(100.0) | 30(100.0) | <0.001 |
| History of diabetes mellitus no. (%) | 3 (10.0) | 3(10.0) | 6(20.0) | 0.421 |
| CKD etiology no. (%)** | | | | <0.001 |
| Chronic glomerulonephritis | 0(0.0) | 18(60.0) | 14(46.7) | |
| Diabetic nephropathy | 0(0.0) | 3(10.0) | 6(20.0) | |
| Hypertensive nephropathy | 0(0.0) | 7(23.3) | 9(30.0) | |
| Other | 0(0.0) | 2(6.7) | 1(3.3) | |
| Creatinine (umol/L) mean(SD)** | 71.4±11.66 | 1179±289.28 | 1104.41±290.44 | <0.001 |
| BUN (mmol/L) mean(SD)** | 5.44±1.32 | 28.83±6.58 | 28.38±7.65 | <0.001 |
| e-GFR (mL/min-1.73 m$^2$)** | 104.42±15.48 | 3.82±0.97 | 4.27±1.33 | <0.001 |
| Duration of dialysis(Month) | 0.00±0.00 | 43.57±25.58 | 46.83±45.32 | 0.732 |
| Hb (g/L) mean(SD)** | 154.37±15.41 | 104.8±18.03 | 108.17±18.48 | <0.001 |
| RBC ($10^{12}$/L) mean(SD) ** | 5.05±0.54 | 3.61±0.68 | 3.64±0.60 | <0.001 |
| ALB (g/L) mean(SD) ** | 45.67±2.80 | 41.66±3.58 | 42.06±5.49 | <0.001 |
| PTH (pg/mL) mean(SD) ** | 38.52±16.03 | 609.71±468.27 | 634.57±657.44 | <0.001 |
| Phosphorus (mmol/L) mean(SD)** | 1.22±0.16 | 2.53±0.62 | 2.4±0.71 | <0.001 |
| Calcium (mmol/L) mean(SD) | 2.34±0.09 | 2.31±0.18 | 2.28±0.20 | 0.317 |
| FGF23 (pg/ml) mean(SD)** | 26.62±23.40 | 96.52±65.07 | 167.26±81.93 | <0.001 |
| D-serine(μM) mean(SD)** | 1.97±0.66 | 9.09±2.71 | 12.27±2.18 | <0.001 |

Notes. NH, Normal hearing; HI, Hearing impairment

** P < 0.01

FGF23 (0<132pg/ml, 1≧132pg/ml), and D-serine (0<10μM, 1≧10μM) as independent variables. In addition, dichotomous logistic regression analysis was performed on the factors influencing the combined hearing impairment in patients with renal failure, and the results showed that rising FGF23 and D-serine were risk factors for hearing impairment in MHD patients (P < 0.01), as detailed in **Table 2**.

### Analysis of factors associated with FGF23

Person correlation analysis of FGF23 with other test parameters was performed on 90 study participants, FGF23 was moderately positively correlated with BUN, blood creatinine, blood potassium, blood phosphorus, PTH, and D-serine, a low positive correlation with UA and a low negative correlation with Hb and RBC (**Table 3**).

### Discussion

Chronic kidney failure and hearing impairment have been treated and studied as separate diseases without much attention to the association between the two diseases. However, the ear and the kidney begin to develop during the same gestation period (5–8 weeks of gestation) in the embryonic stage. Also, the kidney and the cochlea share similarities in anatomy, physiological characteristics, antigenic properties, and even genome [17]. They are also organs with abundant blood supply, high metabolism, and high oxygen consumption, very sensitive to ischemia and hypoxia. Furthermore, there is a correlation between the kidney and the cochlea

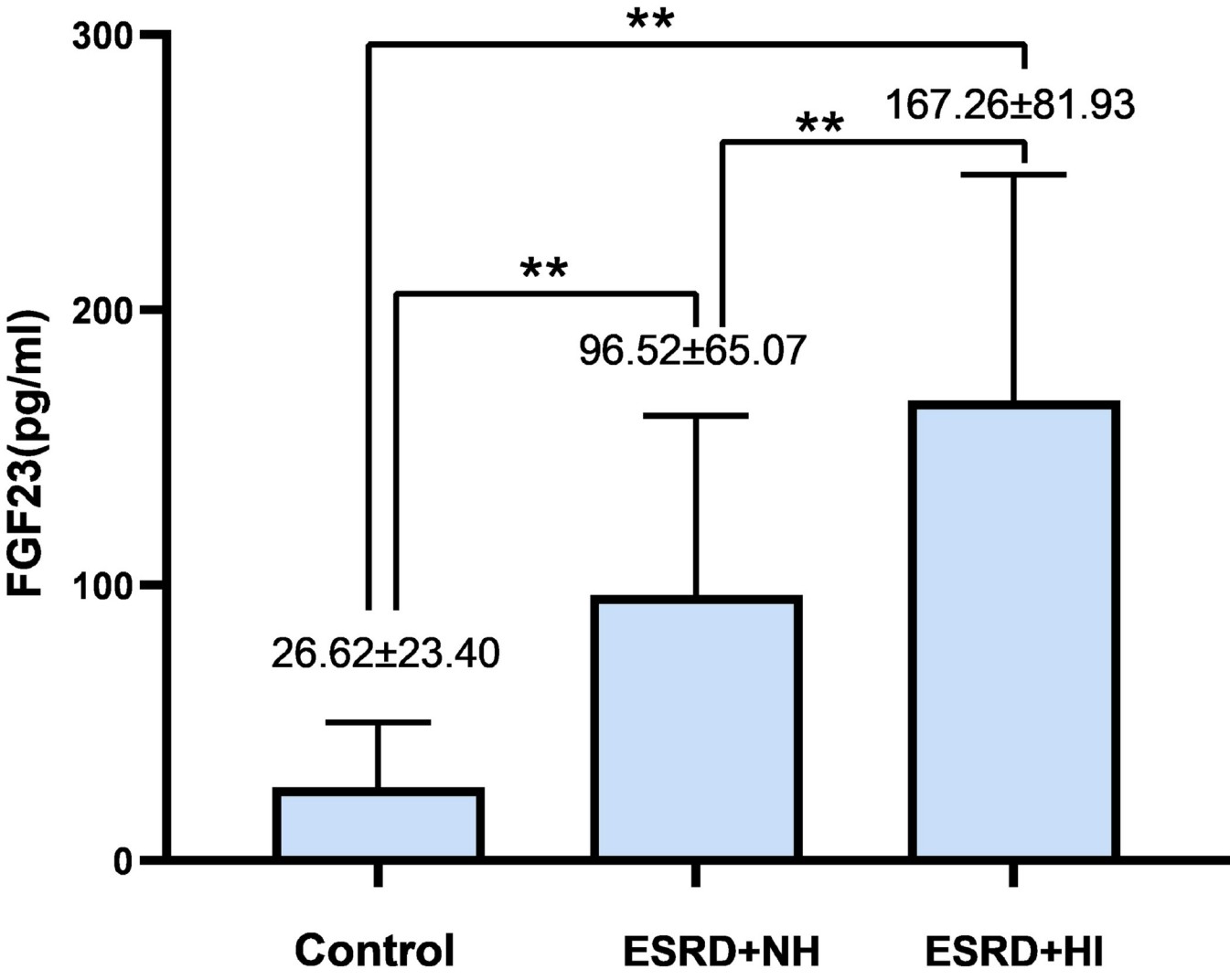

**Fig 2. Comparing the FGF23 levels among 3 groups.**

in terms of enzyme content and distribution, water and electrolyte balance, and pharmacological response to certain drugs (most nephrotoxic drugs are also ototoxic). In addition, the structure of the tubular epithelial cells of the renal tubules is very similar to that of the cells of the vascular rim of the inner ear, and there are similarities between the mechanisms of the inner ear and kidney damage [18]; thus, it is speculated that there is a possibility of "ear-kidney homologation".

In our previous study, we found that the rate of hearing impairment in ESRD patients was significantly higher than in the general population, and the incidence of non-genetic hearing impairment in maintenance hemodialysis patients was as high as 80.5%. Men had significantly higher hearing impairment than women, and a higher proportion of impairment in both ears than in one ear [2]; yet, the exact mechanism of action remained unclear. Furthermore, another study indicated that hearing impairment significantly improved in patients with renal failure after successful renal transplantation when the renal failure was eliminated [19]. Therefore, this study further examined a correlation between ESRD patients and hearing impairment.

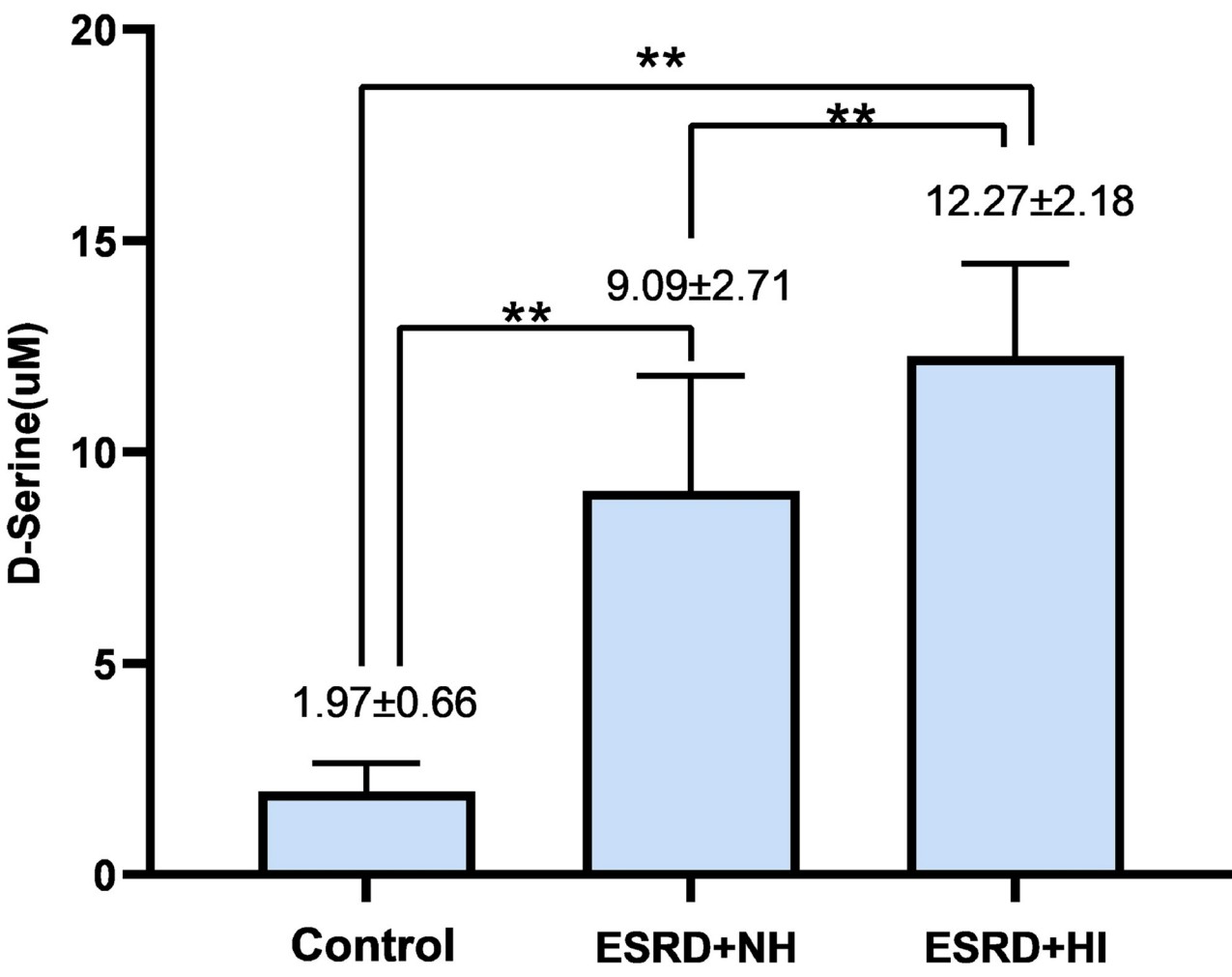

**Fig 3. Comparing the D-serine levels among 3 groups.**

**Table 2. Factors influencing hearing impairment in MHD patients.**

| Variables | B | SE | Wald | P | Exp(B) | 95%CI |
|---|---|---|---|---|---|---|
| Gender | 0.043 | 0.796 | 0.003 | 0.957 | 1.044 | (0.219–4.967) |
| Diabetes mellitus | -0.019 | 1.2 | 0 | 0.988 | 0.981 | (0.093–10.313) |
| Hyperuricemia | 1.047 | 1.003 | 1.091 | 0.296 | 2.85 | (0.399–20.341) |
| Anemia | -0.908 | 0.87 | 1.089 | 0.297 | 0.403 | (0.073–2.22) |
| Hypoalbuminemia | 3.297 | 1.868 | 3.114 | 0.078 | 27.018 | (0.694–1051.512) |
| Potassium | 0.265 | 0.933 | 0.081 | 0.777 | 1.303 | (0.209–8.12) |
| Calcium | 0.121 | 1.128 | 0.012 | 0.914 | 1.129 | (0.124–10.301) |
| Phosphorus | -1.753 | 1.374 | 1.627 | 0.202 | 0.173 | (0.012–2.561) |
| PTH | -0.784 | 0.883 | 0.789 | 0.374 | 0.457 | (0.081–2.576) |
| FGF23 | 2.806 | 0.916 | 9.389 | 0.002 | 16.541 | (2.749–99.549) |
| D-serine | 2.723 | 0.904 | 9.075 | 0.003 | 15.223 | (2.589–89.511) |
| Constant | -8.357 | 4.948 | 2.853 | 0.091 | | |

**Table 3. Analysis of factors associated with FGF23.**

| FGF23 | r | P |
|---|---|---|
| BUN | 0.566 | <0.001 |
| Creatinine | 0.577 | <0.001 |
| UA | 0.333 | 0.001 |
| Hb | -0.391 | <0.001 |
| RBC | -0.392 | <0.001 |
| ALB | -0.119 | 0.262 |
| Potassium | 0.405 | <0.001 |
| Calcium | -0.002 | 0.983 |
| Phosphorus | 0.485 | <0.001 |
| PTH | 0.497 | <0.001 |
| D-serine | 0.683 | <0.001 |

The risk of hearing impairment in MHD patients with abnormally high expression of FGF23 and D-serine was 16.54 and 15.22 times higher than in MHD patients with lower FGF23 and D-serine levels, respectively. Yet, due to the limitations associated with the small sample size, future trials with expanded sample size are needed to further validate the associated risks. In the early stage of renal insufficiency, the body compensates by increasing FGF23 to inhibit the synthesis of $1,25(OH)_2VitD_3$ and promote urinary phosphorus excretion in order to maintain normal blood phosphorus levels. Thus, it is believed that the elevated FGF23 level in the body is a response to the reduced renal phosphorus excretion; yet, when renal function continues to decline and the ability to excrete phosphorus is further reduced, a loss of compensation will occur, i.e., hyper-FGF23 blood and hyperphosphatemia co-exist. Also, high FGF23 levels can cause a variety of mineral and bone metabolism imbalance diseases in the body, such as atherosclerosis, secondary bone disease, calciphylaxis, and cognitive impairment. Previous studies found that FGF23 regulates the immune response and host defense against bacterial infections. Inflammatory cytokines, such as C-reactive protein and IL-6, increase with FGF23 elevation [20]. Inflammatory conditions may also contribute to elevated FGF23 levels in circulation [21]. In addition, diabetes mellitus is associated with inflammation and may lead to diabetic nephropathy. Preclinical studies found that insulin deficiency increases serum FGF23 concentration in mice, a process that can be reversed by administering insulin [22]. Moreover, the elevation of FGF23 can result in inflammation in diabetic nephropathy. Therefore, FGF23 may be an inflammatory marker in diabetic nephropathy, which triggers a series of pathological changes in renal tissue [23].

FGF23 neutralization effectively improves bone quality and osseointegration in CKD mice, suggesting FGF23 as a key factor in CKD-related bone diseases [24]. In addition, the risk of hearing impairment is significantly higher in ESRD patients with higher FGF23 levels than in those with lower FGF23 levels [2], which is consistent with the results of this study. So, FGF23 not only has an important biomarker in the development of CKD but also has a direct pathological effect on its complications.

FGF23 acts through FGFR and α-Klotho. A combination of Klotho/FGFR/FGF23 regulates FGF signal transduction [25]. However, whether a series of pathological changes of FGF23 is essential for Klotho-mediated actions is still under debate [26]. Research shows that Klotho and FGF23 are independently associated with concentric hypertrophy in CKD patients [27]. Our previous study demonstrated that high levels of FGF23 are associated with hearing impairment in ESRD patients, independently of Klotho [2]. So, the Klotho's serum level wasn't evaluated in this study.

D-serine is a physiological co-agonist of the N-methyl-D-aspartate receptor (NMDAR) involved in synaptic plasticity, neurodevelopment, neurodegeneration, and various physiological and pathological activity processes, including excitatory neurotransmission. The conversion of L-serine to D -serine in neurons is facilitated by the action of serine racemase, a pathway called serine shuttle [28]. Overexpression of D-serine mediates the overactivation of NMDAR, causing excitatory neurotoxicity, which leads to neurodevelopmental defects and neurodegeneration. D-serine participates in neurodegeneration and schizophrenia [29, 30]. When D-serine is injected into mice, D-serine rapidly accumulates in the kidney, induces tubular cytotoxicity and contributes to a fibrous phenotype, and accelerates renal remodeling, suggesting that the kidney is an important target organ for D-serine [31] and that D-serine, in addition to being a biomarker, accelerates CKD progression and renal aging, and is a uremic toxin [32]. In addition to being expressed in the central nervous system, D-serine is also expressed in the heart and kidneys and is not only nephrotoxic but also neurotoxic [33].

Our data suggested that abnormally high expression of FGF23 and D-serine increases the risk of developing hearing damage in chronic renal failure. Studies have shown that FGF23 regulates multiple signaling pathways in the body, affecting mineral metabolism, insulin resistance, energy balance, and premature aging [34, 35]. In the renal failure setting, high levels of FGF23 regulate downstream signaling pathways, affecting enzyme activity and ion exchange in the inner ear, damaging cochlear capillaries, leading to cochlear sclerosis, and ultimately affecting normal cochlear function [36]. Meanwhile, the large accumulation of D-serine in chronic renal failure may further lead to hyperfunction of the auditory nerve NMDAR in the brain, exacerbating oxidative stress and excitatory neurotoxicity [37], thus promoting auditory damage.

The present study also found a positive correlation between FGF23 and D-serine in ESRD patients, i.e., as FGF23 increases, D-serine increases accordingly. It is hypothesized that during chronic renal failure, the abnormally high expression of FGF23 activates the downstream signaling pathway and increases the conversion of D-serine through the serine shuttle, which overactivates the NMDAR of the auditory nerve and increases excitatory neurotoxicity, leading to auditory damage. It has also been demonstrated that D-amino acid oxidase (an antagonist of D-serine) reduces endogenous D-serine levels, thereby decreasing NMDAR-mediated neurotoxicity in the hippocampus. Moreover, it was demonstrated that D-serine synergizes with NMDAR to produce neurotoxicity in the brain [38]. Therefore, it is assumed that the FGF23/D-serine axis has a regulatory role in hearing impairment in chronic renal failure and that lowering FGF23 levels may mediate a decrease in D-serine production, ultimately attenuating the effects of excitatory neurotoxicity in the central nervous system on the cochlea and preventing hearing impairment. It provides a potential target for preventing and treating hearing damage in chronic renal failure. In our next study, we plan to use a rat model of hearing impairment in kidney failure to test this hypothesis by targeting the expression of FGF23/D-serine in kidney tissue and cochlea, downstream signaling pathways, and its regulation of NMDAR.

For regular monitoring of hearing status, FGF23 and D-serine levels are needed to assess the risk of hearing impairment in patients with chronic renal failure in order to prevent and detect hearing impairment in a timely manner and to reduce the increased disability, decreased quality of life, and increased disease burden in patients with chronic renal failure due to hearing loss.

## Conclusions

In the present study, we found that FGF23/D-serine levels were significantly higher in MHD patients with hearing impairment than in those with normal hearing and healthy subjects.

Also, FGF23 was positively correlated with D-serine; it is hypothesized that in the chronic renal failure setting, abnormally high expression of FGF23 promotes increased production of D-serine (i.e., the presence of an FGF23/D-serine axis), which overactivates the auditory nerve NMDAR and causes excitatory neurotoxicity, thereby promoting auditory damage. This study provides insights into amino acid imbalance in chronic renal failure and potential biomarkers for early detection of hearing impairment complicated by renal failure. Lowering FGF23/D-serine may provide a new strategy for the pharmacological treatment of hearing impairment complicated by renal failure.

## Study limitations

There are some limitations in the present study. First, it is a single-center study with a small sample size. Therefore, we plan to expand the sample size and perform a multicenter study to reduce the selection bias. Secondly, the pathophysiological regulatory mechanism of FGF23/D-serine in the cochlea and kidney was not fully clarified. In our next study, we plan to conduct animal studies to verify the above hypothesis. Finally, the effect of hemodialysis on hearing in ESRD patients was not included in this study, and the correlation study of this factor of hemodialysis should be addressed by future studies.

## Supporting information

**S1 Data.**
(XLSX)

## Author Contributions

**Conceptualization:** Qing Li.

**Data curation:** Dunlu Yuan, Jiaqing Li, Min Guo, Qing Yang, Jingjing Huang, Ruomei Li.

**Formal analysis:** Jiaqing Li, Min Guo, Jingjing Huang, Qing Li.

**Writing – original draft:** Dunlu Yuan, Jingwen Nie, Qing Li.

**Writing – review & editing:** Dunlu Yuan, Jiaqing Li, Min Guo, Qing Yang, Jingjing Huang, Jingwen Nie, Ruomei Li, Qing Li.

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
