## [Decision Letter · Decision Letter 0]

28 Nov 2022

PONE-D-22-30210Correlation study of FGF23/D-serine in end-stage renal disease patients with combined hearing impairmentPLOS ONE

Dear Dr. Li,

Thank you for submitting your manuscript to PLOS ONE. After careful consideration, we feel that it has merit but does not fully meet PLOS ONE’s publication criteria as it currently stands. Therefore, we invite you to submit a revised version of the manuscript that addresses the points raised during the review process.

We look forward to receiving your revised manuscript.

Kind regards,

Gulali Aktas

Academic Editor

PLOS ONE

Journal Requirements:

Additional Editor Comments:

Authors studied association between FGF23/D-serine in end-stage renal disease patients with hearing loss in the manuscript titled "Correlation study of FGF23/D-serine in end-stage renal disease patients with combined hearing impairment". FGF23 is an inflammatory marker and on the other hand, chronic kidney disease, especially diabetic kidney injury is also characterized with change in serum levels of inflammatory cytokines. I recommend authors discussing this issue.

Reviewers' comments:

Reviewer's Responses to Questions

**Comments to the Author**

1. Is the manuscript technically sound, and do the data support the conclusions?

Reviewer #1: Yes

Reviewer #2: Yes

2. Has the statistical analysis been performed appropriately and rigorously? 

Reviewer #1: Yes

Reviewer #2: Yes

3. Have the authors made all data underlying the findings in their manuscript fully available?

Reviewer #1: Yes

Reviewer #2: Yes

4. Is the manuscript presented in an intelligible fashion and written in standard English?

Reviewer #1: Yes

Reviewer #2: Yes

5. Review Comments to the Author

Reviewer #1: Dear Authors

Thank you very much for the opportunity of reviewing the manuscript entitled ‘Correlation study of FGF23/D-serine in end-stage renal disease patients with combined hearing impairment’ for your journal. The article is about clinically significant FGF23/D-serine on hearing impairment in patients with CRD. Indeed, the article is very interesting and of utmost importance to the field.

Title is appropriate for article, summary abstracted the manuscript adequately, the keyword is enough. The background and the rationale of the study sufficiently expressed in introduction. Study cohort, laboratory and statistical analyses are well issued in methodology. Results of the study are interesting and given in an easy to read way. Discussion is faultless and is the most perfect section of the manuscript. It does not require any more revisions. References are accurate. Figures are clear. Table is informative. As a result, the article should be accepted for publication in the journal.

Sincerely

Reviewer #2: There are a few comments or questions that need clarification from the authors:

1. Why is the title ... in ESRD patients, not CKD patients with routine HD (MHD)? Given that HD plays an important role in phosphate removal, and influences fluctuations in phosphate levels, PTH, and possibly serum FGF-23 levels.

2. OR must be accompanied by 95% CI, because it can describe the accuracy of the data.

3. It is known that FGF-23 plays an important role in the metabolism of vitamin D and phosphate by increasing phosphate excretion with the end result being inhibition of secondary hyperparathyroidism. In this regard, is the presence of FGF-23 serum not just a marker or does it have a direct pathological effect?

4. To be able to bind to its receptor, FGF-23 requires the presence of Klotho. Why wasn't Klotho's serum level evaluated?

5. This study was conducted on patients with GFR < 15 mL/min/1.73 m2. Are all ESRD patients on dialysis?

6. Regarding point 5, it is important to inform the average eGFR and duration of HD in the data on the characteristics of the research subjects.

7. Considering that the HD process can affect the results of laboratory examinations directly or indirectly, the timing of blood sampling is very important related to the HD therapy itself, so further explanation is needed.

8. Similarly, laboratory data is dynamic in nature, whereas hearing loss is a relatively more static condition. Is sampling for laboratory examination carried out once or several times?

6. PLOS authors have the option to publish the peer review history of their article (what does this mean?). If published, this will include your full peer review and any attached files.

Reviewer #1: **Yes: **mehmet zahid kocak

Reviewer #2: No

---

## [Author Response · Author response to Decision Letter 0]

20 Dec 2022

Dear Editor, 

Thank you for carefully reviewing our manuscript previously titled “Correlation study of FGF23/D-serine in maintenance hemodialysis patients with combined hearing impairment” for possible publication in the PLOS ONE. We are grateful to you and your reviewers for their constructive critique. We have revised the manuscript, highlighting our revisions in yellow and have attached point-by-point responses detailing how we have revised the manuscript in response to the reviewers' comments below.

Thank you for your consideration and further review of our manuscript. Please do not hesitate to contact us with any further questions or recommendations.

Yours Sincerely,

Qing Li 

Department of Nephrology, the First Affiliated Hospital of Kunming Medical University, Kunming, 650032, Yunnan, China

Position: Associate chief physician

E-mail: liqing@kmmu.edu.cn；

Tel.: +86-0871-65324888

ORCID of the author: 0000-0001-5452-2734

Reviewer Comments:

Manuscript Number: PONE-D-22-30210

Manuscript Title: Correlation study of FGF23/D-serine in maintenance hemodialysis patients with combined hearing impairment 

Additional Editor Comments 

Authors studied association between FGF23/D-serine in end-stage renal disease patients with hearing loss in the manuscript titled "Correlation study of FGF23/D-serine in end-stage renal disease patients with combined hearing impairment". FGF23 is an inflammatory marker and on the other hand, chronic kidney disease, especially diabetic kidney injury is also characterized with change in serum levels of inflammatory cytokines. I recommend authors discussing this issue. 

Response:

Done as what you suggested. We deeply appreciate your good suggestion. We have added discussion about this issue. Please see Discussion section: Page 15, Line 232-242, References section: Page23, Line 414-424.

Previous studies found that FGF23 regulates the immune response and host defense against bacterial infections. Inflammatory cytokines, such as C-reactive protein and IL-6, increase with FGF23 elevation(20). Inflammatory conditions may also contribute to elevated FGF23 levels in circulation(21). In addition, diabetes mellitus is associated with inflammation and may lead to diabetic nephropathy. Preclinical studies found that insulin deficiency increases serum FGF23 concentration in mice, a process that can be reversed by administering insulin(22). Moreover, the elevation of FGF23 can result in inflammation in diabetic nephropathy. Therefore, FGF23 may be an inflammatory marker in diabetic nephropathy, which triggers a series of pathological changes in renal tissue (23).

References

20. Zheng Z, Zheng F. Immune cells and inflammation in diabetic nephropathy. J Diabetes Res. 2016;2016:1841690.

21. Rossaint J, Unruh M, Zarbock A. Fibroblast growth factor 23 actions in inflammation: a key factor in CKD outcomes. Nephrol Dial Transplant.2017; 32(9):1448-53.

22. Bär L,Feger M,Fajol A.Insulin suppresses the production of fibroblast growth factor 23 (FGF23).Proceedings of the national academy of sciences of the United States of America.2018;115 (22):5804-9.

23. David V, Martin A, Isakova T, Spaulding C, Qi L, Ramirez V, et al. Inflammation and functional iron deficiency regulate fibroblast growth factor 23 production. Kidney Int. 2016;89(1):135-46.

Reviewer 2

1. Why is the title ... in ESRD patients, not CKD patients with routine HD (MHD)? Given that HD plays an important role in phosphate removal, and influences fluctuations in phosphate levels, PTH, and possibly serum FGF-23 levels. 

Response:

Done as what you suggested. We deeply appreciate your good suggestion. We have corrected the title: Correlation study of FGF23/D-serine in maintenance hemodialysis patients with combined hearing impairment

Please see the title section: Page 1, Line 1-3.

Title: Correlation study of FGF23/D-serine in maintenance hemodialysis patients with combined hearing impairment

2. OR must be accompanied by 95% CI, because it can describe the accuracy of the data.

Response:

We deeply appreciate your good suggestion. We have added 95%CI. 

Please see Abstract section: Page 2, Line40-41.

Also, elevated FGF23 and D-serine were identified as risk factors for hearing impairment in ESRD, with ORs of 16.54 (95%CI, 2.75-99.55) and 15.22 (95% CI,  2.59-89.51), respectively.

3. It is known that FGF-23 plays an important role in the metabolism of vitamin D and phosphate by increasing phosphate excretion with the end result being inhibition of secondary hyperparathyroidism. In this regard, is the presence of FGF-23 serum not just a marker or does it have a direct pathological effect?

Response:

We deeply appreciate your good suggestion. We have added the discussion.

Please see Introduction section: Page4-5, Line69-74；Discussion section: Page15, Line 243-249，References：Line 425-427.

.

Thus, elevated FGF23 is considered a sensitive biomarker for renal and extrarenal adverse effects in patients with CKD (10). Furthermore, abnormally high expression of FGF23 in patients with chronic renal failure accelerates a series of extrarenal damages such as organismal calciphylaxis, atherosclerosis, secondary bone disease, and cognitive impairment of the nervous system.

FGF23 neutralization effectively improves bone quality and osseointegration in CKD mice, suggesting FGF23 as a key factor in CKD-related bone diseases (24). In addition, the risk of hearing impairment is significantly higher in ESRD patients with higher FGF23 levels than in those with lower FGF23 levels (2), which is consistent with the results of this study. So, FGF23 not only has an important biomarker in the development of CKD but also has a direct pathological effect on its complications.

References

24.Sun N, Guo Y, Liu W, Densmore M, Shalhoub V, Erben RG , et al.FGF23 neutralization improves bone quality and osseointegration of titanium implants in chronic kidney disease mice. Scientific reports. 2015; 5:8304. 

4. To be able to bind to its receptor, FGF-23 requires the presence of Klotho. Why wasn't Klotho's serum level evaluated?

Response:

We deeply appreciate your good suggestion. We have added the explanation.

Please see Discussion section: Page15, Line 250-257. References：Line 429-438.

FGF23 acts through FGFR and α-Klotho. A combination of Klotho/FGFR/FGF23 regulates FGF signal transduction (25). However, whether a series of pathological changes of FGF23 is essential for Klotho-mediated actions is still under debate (26). Research shows that Klotho and FGF23 are independently associated with concentric hypertrophy in CKD patients (27). Our previous study demonstrated that high levels of FGF23 are associated with hearing impairment in ESRD patients, independently of Klotho (2). So, the Klotho's serum level wasn’t evaluated in this study.

References

25. Muñoz-CJR,Rodelo HC, Pendon RMV, Martin MA, Santamaria R, Rodriguez M; Klotho/FGF23 and Wnt Signaling as Important Players in the Comorbidities Associated with Chronic Kidney Disease .Toxins(Basel) , 2020; 12(3): 185. 

26.Tanaka S, Fujita S, Kizawa S, Morita H, Ishizaka N. Association between FGF23, α-Klotho, and Cardiac Abnormalities among Patients with Various Chronic Kidney Disease Stages. PLoS ONE , 2016;11, e0156860.

27. Silva AP,Mendes F,Carias E, Gonçalves RB, Fragoso A, Dias C, et al.Plasmatic Klotho and FGF23 Levels as Biomarkers of CKD-Associated Cardiac Disease in Type 2 Diabetic Patients. International journal of molecular sciences,2019;20 (7) :1536. 

5. This study was conducted on patients with GFR < 15 mL/min/1.73 m2. Are all ESRD patients on dialysis?

Response:

Yes，all ESRD patients on dialysis in the present study.

Please see Material and Methods section: Page 6, Line98-100. 

Inclusion criteria were: age ≥ 18 years; glomerular filtration rate (GFR) < 15 mL/min-1.73m2; receiving regular hemodialysis for more than 3 months, four hours per session, 3 times/week

6. Regarding point 5, it is important to inform the average eGFR and duration of HD in the data on the characteristics of the research subjects.

Response:

Done as what you suggested. We deeply appreciate your good suggestion. We have added the average eGFR and duration of HD in the data on the characteristics of the research subjects.

Please see Material and Methods section: Page 6, Line98-100, Results section: Page 10-11, Table 1. 

Inclusion criteria were: age ≥ 18 years; glomerular filtration rate (GFR) < 15 mL/min-1.73 m2; receiving regular hemodialysis for more than 3 months, four hours per session, 3 times/week; conscious; willing to participate in the study.

Characteristic Control group (n=30) MHD+NH 

group

(n=30) MHD+HI 

group

(n=30) P

Age(y) mean(SD) 43.17±8.95 43.43±8.79 43.90±10.38 0.954

Male no. (%) 16(53.3) 15(50.0) 22(73.3) 0.139

BMI (Kg/m2) mean(SD) 23.96±3.19 22.30±3.90 22.53±3.61 0.156

History of hypertension no. (%)** 12(40.0) 30(100.0) 30(100.0) ＜0.001

History of diabetes mellitus no. (%) 3 (10.0) 3(10.0) 6(20.0) 0.421

CKD etiology no. (%)** ＜0.001

Chronic glomerulonephritis 0(0.0) 18(60.0) 14(46.7) 

Diabetic nephropathy 0(0.0) 3(10.0) 6(20.0) 

Hypertensive nephropathy 0(0.0) 7(23.3) 9(30.0) 

Other 0(0.0) 2(6.7) 1（3.3） 

Creatinine (umol/L) mean(SD)** 71.4±11.66 1179±289.28 1104.41±290.44 ＜0.001

BUN (mmol/L) mean(SD)** 5.44±1.32 28.83±6.58 28.38±7.65 ＜0.001

e-GFR（mL/min-1.73 m2)** 104.42±15.48 3.82±0.97 4.27±1.33 ＜0.001

Duration of dialysis(Month) 0.00±0.00 43.57±25.58 46.83±45.32 0.732

Hb (g/L) mean(SD)** 154.37±15.41 104.8±18.03 108.17±18.48 ＜0.001

RBC (1012/L) mean(SD) ** 5.05±0.54 3.61±0.68 3.64±0.60 ＜0.001

ALB (g/L) mean(SD) ** 45.67±2.80 41.66±3.58 42.06±5.49 ＜0.001

PTH (pg/mL) mean(SD) ** 38.52±16.03 609.71±468.27 634.57±657.44 ＜0.001

Phosphorus (mmol/L) mean(SD)** 1.22±0.16 2.53±0.62 2.4±0.71 ＜0.001

Calcium (mmol/L) mean(SD) 2.34±0.09 2.31±0.18 2.28±0.20 0.317

FGF23 (pg/ml) mean(SD)** 26.62±23.40 96.52±65.07 167.26±81.93 ＜0.001

D-serine(μM）mean(SD)** 1.97±0.66 9.09±2.71 12.27±2.18 ＜0.001

Notes. NH, Normal hearing ; HI, Hearing impairment ; ** P < 0.01

7. Considering that the HD process can affect the results of laboratory examinations directly or indirectly, the timing of blood sampling is very important related to the HD therapy itself, so further explanation is needed.

Response:

Done as what you suggested. We deeply appreciate your good suggestion. We have added the explanation. 

Please see Data Collection section: Page 7, Line134-137.

All subjects fasted for at least 8 h. Then, 5 ml of venous blood was collected before 8:30 am. To avoid the influence of hemodialysis on laboratory data in MHD patients, blood was not collected during and 1 day after hemodialysis. Sampling for laboratory examination was carried out one-time. 

8. Similarly, laboratory data is dynamic in nature, whereas hearing loss is a relatively more static condition. Is sampling for laboratory examination carried out once or several times? 

Response:

We deeply appreciate your good suggestion. We have added the explanation. 

Please see Data Collection section: Page 7, Line135-137.

To avoid the influence of hemodialysis on laboratory data in MHD patients, blood was not collected during and 1 day after hemodialysis.Sampling for laboratory examination was carried out one time. 

We deeply appreciate your constructive comments.

---

## [Decision Letter · Decision Letter 1]

28 Dec 2022

Correlation study of FGF23/D-serine in maintenance hemodialysis patients with combined hearing impairment

PONE-D-22-30210R1

Dear Dr. Li,

We’re pleased to inform you that your manuscript has been judged scientifically suitable for publication and will be formally accepted for publication once it meets all outstanding technical requirements.

Kind regards,

Gulali Aktas

Academic Editor

PLOS ONE

Additional Editor Comments (optional):

Authors successfully revised the paper in accordance with the reviewers' suggestions.

Reviewers' comments:

Reviewer's Responses to Questions

**Comments to the Author**

1. If the authors have adequately addressed your comments raised in a previous round of review and you feel that this manuscript is now acceptable for publication, you may indicate that here to bypass the “Comments to the Author” section, enter your conflict of interest statement in the “Confidential to Editor” section, and submit your "Accept" recommendation.

Reviewer #1: All comments have been addressed

Reviewer #2: All comments have been addressed

2. Is the manuscript technically sound, and do the data support the conclusions?

Reviewer #1: Yes

Reviewer #2: (No Response)

3. Has the statistical analysis been performed appropriately and rigorously? 

Reviewer #1: No

Reviewer #2: (No Response)

4. Have the authors made all data underlying the findings in their manuscript fully available?

Reviewer #1: Yes

Reviewer #2: (No Response)

5. Is the manuscript presented in an intelligible fashion and written in standard English?

Reviewer #1: Yes

Reviewer #2: (No Response)

6. Review Comments to the Author

Reviewer #1: (No Response)

Reviewer #2: (No Response)

7. PLOS authors have the option to publish the peer review history of their article (what does this mean?). If published, this will include your full peer review and any attached files.

Reviewer #1: No

Reviewer #2: No

---

## [Editor Report · Acceptance letter]

6 Jan 2023

PONE-D-22-30210R1 

Correlation study of FGF23/D-serine in maintenance hemodialysis patients with combined hearing impairment 

Dear Dr. Li:

I'm pleased to inform you that your manuscript has been deemed suitable for publication in PLOS ONE. Congratulations! Your manuscript is now with our production department. 

Kind regards, 

on behalf of

Professor Gulali Aktas 

Academic Editor

PLOS ONE